# Academic User View: Organ-on-a-Chip Technology

**DOI:** 10.3390/bios12020126

**Published:** 2022-02-16

**Authors:** Mathias Busek, Aleksandra Aizenshtadt, Mikel Amirola-Martinez, Ludivine Delon, Stefan Krauss

**Affiliations:** 1Hybrid Technology Hub—Centre of Excellence, Institute of Basic Medical Sciences, University of Oslo, 0317 Oslo, Norway; mathias.busek@medisin.uio.no (M.B.); aleksandra.aizenshtadt@medisin.uio.no (A.A.); m.a.martinez@medisin.uio.no (M.A.-M.); 2Department of Immunology and Transfusion Medicine, Oslo University Hospital, 0424 Oslo, Norway; ludivine.delon@medisin.uio.no

**Keywords:** organ-on-a-chip (OoC), microphysiological systems (MPS), survey, usability, limitations, disease models, drug testing

## Abstract

Organ-on-a-Chip (OoC) systems bring together cell biology, engineering, and material science for creating systems that recapitulate the in vivo microenvironment of tissues and organs. The versatility of OoC systems enables in vitro models for studying physiological processes, drug development, and testing in both academia and industry. This paper evaluates current platforms from the academic end-user perspective, elaborating on usability, complexity, and robustness. We surveyed 187 peers in 35 countries and grouped the responses according to preliminary knowledge and the source of the OoC systems that are used. The survey clearly shows that current commercial OoC platforms provide a substantial level of robustness and usability—which is also indicated by an increasing adaptation of the pharmaceutical industry—but a lack of complexity can challenge their use as a predictive platform. Self-made systems, on the other hand, are less robust and standardized but provide the opportunity to develop customized and more complex models, which are often needed for human disease modeling. This perspective serves as a guide for researchers in the OoC field and encourages the development of next-generation OoCs.

## 1. Introduction

Organ-on-a-Chip (OoC) models combine microsystem technology, cell biology, and microfluidics to mimic organ functionality in vitro. They are on the verge of widespread use as new models for studying diseases and drug response/toxicity in both academia and the pharmaceutical industry [1]. A variety of platforms were developed from 2010 onwards when the first complex OoC, a lung-on-a-chip model, comprising a co-cultivation of lung alveolar and capillary cells on the two sides of a porous membrane, was presented [2]. Most of the commercial OoC platforms on the market to date address the needs of the pharmaceutical industry. Hence, several studies have been conducted to analyze the industry’s requirements [3] for advancing OoC technology towards predictive drug screening and regulatory approval [4]. Until now, similar need projections for academic users have not been conducted. Therefore, we surveyed peers both in academia and industry to identify advantages, current limitations, and desired future features in OoC technology.

## 2. Overview of Organ-on-Chip Technology

The application of microfluidic systems in biological research has more than 20 years of history, dating back to the process of soft lithography using transparent and gas-permeable polydimethylsiloxane (PDMS) [5]. From 2010 onward, advanced combinations of microsystem technology and cell biology, including 3D models (organoids, bioprinting), gave rise to a whole new field—organ-on-a-chip (OoC)—which is constantly expanding (>1800 original research articles and >1000 reviews over the past 10 years, as extracted from EMBASE, PubMed, Scopus, and Web of Science). Microfluidic devices integrating in vitro cellular constructs are also often termed microphysiological systems which reflects the key aim of OoCs—the recapitulation of physiological processes in multi-cellular systems. As such, OoCs may integrate essential factors of the cellular microenvironment, such as multiple cell types, vascularization, immune competence, barrier functions, mechanical stimulation (e.g., shear force), and gradients of biomolecules in vitro. The numerous possibilities with OoC customization enable researchers not only to choose the complexity level of the cellular structures (non-3D and 3D) and media perfusion (gravity [6] or external/internal pump [7]) but also to adapt the system to a plethora of readouts (Figure 1). 

Based on the above features, OoC models are expected to bridge the so-called “translational gap” between (patho-) physiological processes in conventional in vitro and animal models on one side and in humans on the other side [8]. Therefore, OoC technology is projected to be able to reduce research and development costs in the pharmaceutical industry by 10–26% [9]. Accordingly, substantial progress has been achieved in the development of standardized systems suitable for high throughput tests in industrial settings [10,11] and a large number of companies that are developing and working with OoC systems have emerged. 

These companies offer ready-to-use devices, including operating systems or/and contract research services. Table 1 gives an overview of companies working in the OoC arena that are identifiable by PubMed and LinkedIn searches. Based on these criteria, we counted more than 30 companies developing OoC systems, excluding emerging OoC startups and microfluidic companies providing only devices/equipment for operation but no cell or organ models.

An analysis of organ and tissue models of these companies identifies liver, lung, gut, kidney, and heart in the pipeline of most companies, illustrating the early focus of OoCs on preclinical safety testing. On the contrary, very few of them developed OoCs for islets, muscle tissue, cartilage, and bone marrow.

We also found that, according to the available information, only a few companies integrate tissue vascularization, whereby we define vascularization as a connection of the tissue or organ model to a network of functional microvascular vessels [12]. Many OoCs include immune cells under static conditions, which we do not list as “circulating cells” in the graph. However, we only list models providing a flow-based recirculation of cells, 

Moreover, the analysis shows that OoC companies are trying to make their platforms more user-friendly by using internal pumps, automatized external pumps, or gravity-driven flow for perfusion (see Figure 1C) to provide “full-stop solutions” for chip preparation (flow, stretch, oxygen, bio-molecular gradients, temperature), operation, and readouts (e.g., Zoë-CM2™ by Emulate, PREDICT96 by Draper Labs, PhysioMimix™ by CN-Bio Innovations, OrganoTEER^®^ platform from Mimetas) [13,14,15].

An important driver for OoC development is the 3R principle (replace, reduce, and refine animal testing), promoted by regulatory bodies. Examples are the Innovation Task Force (ITF) [16] from the European Medicines Association (EMA) and the Alternative Methods Working Group from the US Food and Drug Administration (FDA) [17], focusing on advancing preclinical models by the 3R principles. However, OoCs have so far not been included in the regulatory guidelines for drug approval [11]. A necessary condition for the full implementation of OoCs in the pipeline for drug testing (both safety and efficacy) is the fulfillment of the following criteria: validity, relevance, reliability, and sensitivity. Nevertheless, recently OoCs reached a stage of maturity to influence internal decision making in pharmaceutical companies during drug discovery/testing studies [4]. 

The willingness of stakeholders of the pharmaceutical industry to include OoC technology in preclinical studies is therefore clearly visible [10]. However, except for a few cases of well-established assays (blood vessel, gut, liver, and lung models) from first-generation companies (CNBio, Mimetas, Emulate, TissUse), OoCs are still not broadly adopted and are far from being able to replace or even reduce animal models [10].

While the industry focuses on relevance, simplicity, reproducibility, and scalability at a competitive cost, academic research groups are the main drivers towards more complex and more diverse physiological and disease-relevant OoC platforms [18,19,20]. Increased complexity, however, comes along with reduced usability and the need for more hardware to operate the devices. This generates a significant entry barrier for end users to use academic platforms and affects both transferability and throughput [21]. Moreover, there is a need for standardization of OoC design, biological material, growth conditions, and readout methods [11]. To help in this process, the National Institute of Health (NIH) has set up “Tissue Chip Testing Centers” [22], that try to compare results obtained on different commercial and academic platforms.

OoC technology allows researchers to not only model physiological and disease processes on-chip, but also allows monitoring and tracking of these processes in real time (Figure 1D). The currently most common readout methods are optical, including bright-field microscopy, fluorescence microscopy, and post-fixation confocal microscopy, while Raman confocal microscopy [23] is emerging. Many OoC studies include advanced analytical “omics” methods, including scRNA sequencing, proteomics, and metabolomics. Besides these general technologies, some OoC platforms integrated sensors for selected readouts (e.g., oxygen, specific metabolites, cell toxicity, barrier permeability, and electrical signals) [24,25]. Considering the high amount of data generated by both high-throughput screening and advanced readout methods, there is an increasing need to use machine learning in data analysis [26].

In the light of the rapidly progressing field, we asked scientists in the OoC arena for the specific advantages and shortcomings of existing OoC platforms and desired features for next-generation OoC solutions.

## 3. User View Survey

Using a dedicated survey scheme (Appendix A), we collected 187 answers from academia (146) and industry (41) in 35 countries (Dataset in Appendix A). An overview of the research fields, positions, and countries of the participants is given in Appendix A. The survey was distributed via international networks including the European OoC society, the Nordic OoC network, relevant Facebook groups [27], LinkedIn, and collaborators at different universities (a list of asked institutions is given in Appendix A).

We split the responses into two groups: (A) researchers currently using OoC (51.3%) and (B) researchers not using OoC technology yet (48.7%). The active OoC users were further divided into (A-I) researchers using self-made devices or devices designed in collaborative work (60), (A-II) researchers using commercial systems (11), and (A-III) researchers using both (14). OoC platform users were first asked for the main advantages/drawbacks of their current platforms (Figure 2) and the degree of satisfaction with their OoC solution. In group A-I, users gave their systems an overall rating of 6.08 ± 1.8 points (1 to 10) on average, while groups A-II and A-III gave an average rating of 7.9 ± 1.5 and 6.9 ± 1.8, respectively.

To compare responses from the different user groups and to rate the multiple answer options, counts per answer were normalized to the number of responses in this group.

### 3.1. Advantages and Drawbacks of Current OoC Platforms

Although the different OoC user groups rated the currently used platforms quite equally, self-made and commercial devices were reported to have a different set of limitations and advantages (Figure 2). Commercial platforms scored high on usability, being well established in the laboratory and easy to use, while many users requested a higher complexity than provided by these platforms. Self-made devices, on the other hand, were referred to as “established in our lab” in the figures. The self-made platforms also seem to frequently suffer from common problems, such as air bubbles, but provide a higher level of complexity compared with commercial platforms. Users with experience in both commercial and self-made systems marked low long-term stability, high prices, and insufficient model validation as major drawbacks. The listed answers highlight an imbalance between commercial, robust, and easy-to-use systems on one side and the necessity for increased complexity and better in vitro models on the other side.

### 3.2. Obstacles for Broader Usage of Current OoC Platforms

From the group of scientists that do not work with OoC platforms, the majority (55.6%) of respondents indicated an interest in the technology but highlighted several concerns such as unavailability of ready-to-use devices, complicated use (high entry barrier), unavailability of production facilities, and high costs (Figure 3). The rest of the participants were unsure (37.8%) if they want to use OoC in research, whereas only 6.7% were not interested at all.

### 3.3. Desired Features for OoC Technology

Next, participants were asked about desired features for OoC systems (Figure 4). The majority of respondents from all groups see readout and imaging, followed by vascularization, as important, must-have features. Vascularization is a central structural element in advanced organ representations and an important step towards a better physiological relevance of in vitro models. Interestingly, the immune system—which is a key element in numerous disease models—featured less prominently on the wish list of necessary features. It probably reflects the complexity of the immune systems and, hence, the feasibility issue. Functions such as oxygen concentration, growth factor gradients, and mechanical stimuli were featured on the wish list of all responders. The interactions of several organs scored high on the wish list of researchers that do not have “hands-on” OoC experience yet (group B), but are remarkably low on the list of active OoC users (group A), most likely reflecting the significant challenges of multi-OoCs. Finally, OoC platforms for testing drug response featured robustly in all groups of participants, pointing towards a broad interest in drug discovery and testing in the OoC field.

## 4. Discussion

The OoC market is increasing; in 2020, it was evaluated to have a volume of USD 9.8 mill. and is expected to grow to USD 178.6 mill. in 2030 only in North America [28]. A plethora of commercial platforms has been developed in recent years as highlighted by Table 1. Nevertheless, it seems that many academic users are still developing their own OoC systems as they cannot find a platform that fits to their application. One possible reason for this might be that many OoC companies mainly focus on the pharmaceutical industry as main customers and their platforms do not sufficiently address the broad span of academic exploration and needs. Another reason may be that there are a number of academic environments that move into the OoC arena from engineering and material science disciplines, exploring new materials and platform designs, while adding cell biology competence. New OoC platforms should address the shortfalls of existing commercial and non-commercial systems by allowing a sufficient level of complexity combined with robustness and standardization, while providing manageable ease of use. Platforms that are compatible with both simple routine readout and imaging systems and more advanced analysis may also be considered as a central necessity in OoC technology.

The average user rating of commercial OoC platforms was rated highest and the satisfaction was lowest among users of self-made systems. This could also be underpinned by the list of problems users encountered when using the platforms (Figure 2b). It seems that many difficulties, such as low robustness, have been already optimized in commercial platforms, thus leading to a generally higher success rate when working with the systems.

In our survey, we also included feedback from industrial users. Where the feedback of both groups was quite similar in many points, we could identify significant differences in what both user groups struggle with (Appendix A). It seems that industrial users do not struggle a lot with air bubbles and robustness, but see the limited biological complexity as the weakest point. Interestingly, from the industrial users we asked, 53.8% are using commercial OoC platforms or both self-made and commercial systems, whereas only 21.1% among academic users include commercial platforms in their studies. This fact again underpins the finding that commercial OoC platforms are not so common among academic users and seem to address the needs of industrial (pharmaceutical) users to a larger extent. Furthermore, if one considers the literature about end-user perspectives on OoC technology [3,4,29], it becomes clear that most commercial platforms are tailored for pharmaceutical testing. Only a few commercial platforms are suitable e.g. for disease modelling and thus of interest for academic users.

An important aspect highlighted by our survey is the technological entry barrier in design and fabrication of OoC devices (Figure 3). Furthermore, OoC platforms have to be tailored to biological questions and needs, requiring a substantial level of interdisciplinarity. Commonly OoC models are developed based on a biological/medical question. From that, a concept is developed, possible designs are modeled, an OoC system is fabricated, and the biological model is established. Therefore, interdisciplinary training networks, e.g., from the EU [30], have been launched to qualify Ph.D. students in both biology and microfabrication/engineering.

Our survey suggests that OoC companies may profit from more actively assisting academic researchers to overcome entry barriers in the use of OoC platforms and/or production of their OoC platforms. The development of OoC platforms that target the specific needs of academic users has the potential to lead to joined projects with biotechnology companies and the pharmaceutical industry. Based on recent progress in OoC technology, we anticipate that incentives for collaborations between pharmaceutical industries, OoC producers, and academic researchers will enhance faster development of the next-generation OoC models, bridging commercial and academic needs and advancing standardization and reproducibility of new and more complex OoC platforms.

Our survey identified specific needs for the user groups of commercial and non-commercial OoC platforms. Imaging and readout methods rank highest in the need projections for all OoC systems (Figure 4), highlighting the main benefit of this technology as transparent chip materials allow real-time tracking of a biological specimen by various methods.

The second feature considered to be important for all platforms is the ability to test drug response. While the effect of drugs may differ between the sources and states of the biological material and hence offer a possibility for personalized medicine, it is also influenced by the OoC system itself. Still, OoCs are expected to be closer to the human drug response than animal models.

Clear differences in need projections can be seen for instance for the aspect of “organ interaction”. Users of commercial systems highlight this feature, whereas users of non-commercial platforms focus more on details that impact organ physiology and complexity including oxygen/nutrient gradients, mechanical stimulation, and immune interaction in one type of tissue or organ model. These aspects are not of particular interest to users of commercial systems, which is in line with the availability of systems. For drug screening, for instance, organ interactions are often important; therefore, most commercial platforms focus especially on this aspect either by implementing media recirculation to different organ compartments [31] or by using pipetting robots to couple single organ models with each other [32].

Mechanical stimulation is important for cell functionality and maturation [18]—most prominently by flow-induced shear stress [33]. Specifically designed OoC systems are required to model shear stress, which explains why this aspect is mainly highlighted by the users of self-made systems. 

Another important aspect highlighted by the participants was vascularization. While some commercial systems are designed to enable vascularization [12], most commercial systems lack this important functionality. A possible reason for this is that membrane-based models, a standard design of first-generation OoC platforms (Figure 1), do commonly not allow endothelial outgrowth. Only real 3D systems (Figure 1) with organ models embedded in an extracellular matrix allow for generation of a microvascular network.

Finally, immune-competent chips are an emerging topic [34], often requiring highly specialized OoC designs. Hence, immune competence may be an OoC functionality that projects further in the future and will have to be pioneered in the academic arena before it may be implemented in commercial OoC designs.

## Figures and Tables

**Figure 1 biosensors-12-00126-f001:**
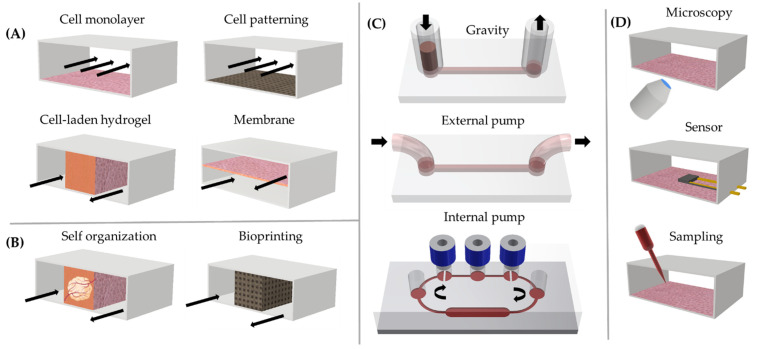
Main components of current OoC systems: (**A**) non-3D models, (**B**) 3D models, (**C**) perfusion modes, and (**D**) readouts.

**Figure 2 biosensors-12-00126-f002:**
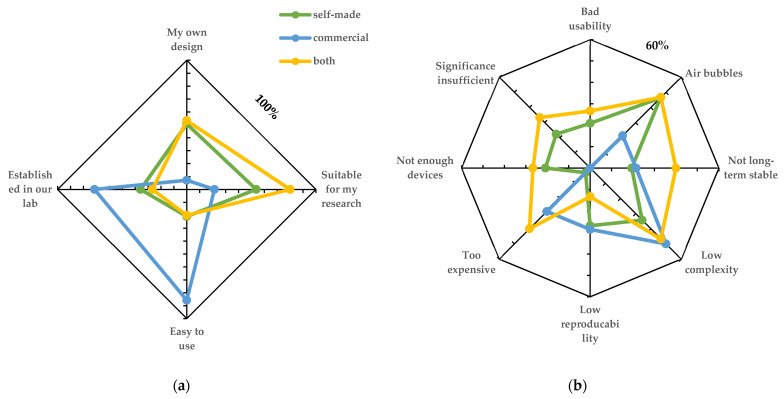
Assessment from researchers currently using OoC technology (group A) regarding (**a**) advantages and (**b**) drawbacks of their current platforms.

**Figure 3 biosensors-12-00126-f003:**
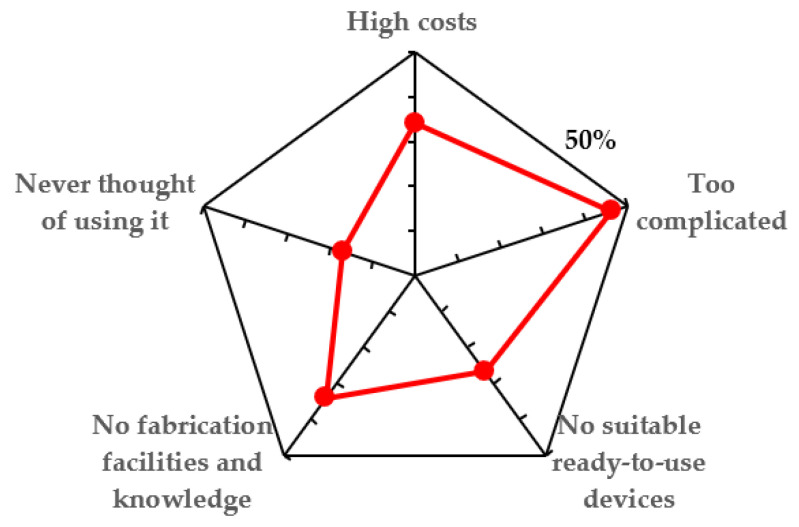
Obstacles for OoC usage from researchers not using OoC technology yet (group B).

**Figure 4 biosensors-12-00126-f004:**
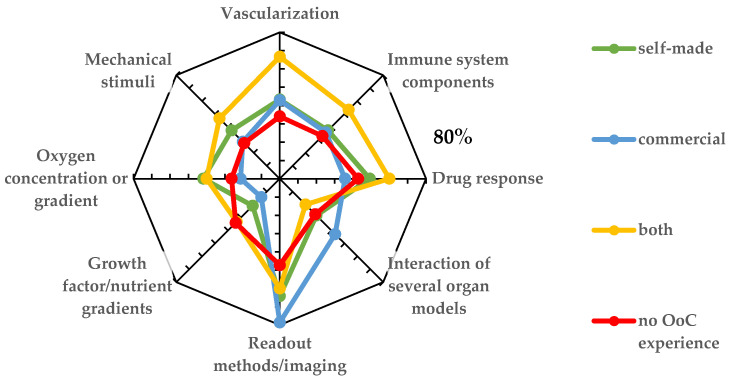
Desired features of OoC platforms are sorted by users of self-made and commercial platforms as well as by researchers with no OoC experience.

**Table 1 biosensors-12-00126-t001:** Overview of OoC companies.

Name of Company	Organs/Tissue Models																	
	Brain/ neurons	Lung	Liver	Gut	Kidney	Islet	Muscles	Heart	Skin	Cartilage	Bone Marrow	Microvasculature	Circulating Cells	Organ Interaction	One-stop Solution	External Pump	Numbers of Publications	Year of Foundation
Aim biotech																	75	2012
Altis BioSystems																	18	2015
Ananda Devices																	1	2015
Alveolix																	18	2019
Aracari Bio																	10	2019
Axosim																	9	2014
Beonchip																	3	2016
Biomimx																	5	2017
BI/OND																	3	2017
CNBio																	24	2009
Dynamic42																	12	2018
EHT Technologies																	62	2015
Draper (PREDICT-96)																	5	2019
Emulate																	20	2014
Hesperos																	45	2015
Ibidi GmbH																	>100	2001
InSphero																	14	2009
Jiksak Bioengineering																	3	2017
Kirkstall																	16	2006
MesoBioTech																	2	2016
MicroBrainBT																	4	2014
Mimetas																	61	2013
Nortis BIO																	20	2012
REVIVO Biosystems																	0	2019
SynVIVO																	40	2015
Tara biosystems																	20	2014
TissUse																	60	2010
Xona microfluidics																	170	2008

## Data Availability

Data used for this study can be found in the Appendix A.

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
