# Peer review of "Academic User View: Organ-on-a-Chip Technology"

_biosensors, 2022, doi:10.3390/bios12020126_

Round 1

Reviewer 1 Report

The paper presents the survey of organ-on-a-chip users about their attitude and feeling of such devices and platforms. The paper provides novel information that is describing the development of the field. It is well written and the presentation is clear. 

The main issue of the paper is that in the survey very few people were using commercial (13%) and both commercial and custom (16%) types of devices. The most of the questioned people were using only custom devices. Therefore there is a contradiction - in the Table 1 there are mentioned 24 companies that produce OoC devices but very few questioned people are using them. Could you provide approximate market volume of the commercial devices to get the idea how many of them are currently being used?

Also there were few people from industry taking part in the survey (13%). Do you have representative sample of commercial systems users? What types of OoC devices use users from industry?

Overall, the paper might be accepted after answering these questions.

Reviewer 2 Report

This work deserves publication as the presented information can be of great importance for the future development and valorisation of organ-on-a-chip models. To further strengthen this, I would ask the authors to include 2 additional types of information:

  • To indicate the geographical origin of the participants of the survey such that the reader can evaluate if there is a diffused spread among the participants.
  • To include a table in the supplementary material with the names of the institutes that participated in the survey. This will enable potential collaborations between the survey participants and the readers (who for example might have good fabrication facilities). Obviously, I am fully aware that the survey must be kept anonymous but a general listing of the institutes without notifying individual answers can already be of great help.
